# Phage-Derived Depolymerase: Its Possible Role for Secondary Bacterial Infections in COVID-19 Patients

**DOI:** 10.3390/microorganisms11020424

**Published:** 2023-02-07

**Authors:** Amina Nazir, Jiaoyang Song, Yibao Chen, Yuqing Liu

**Affiliations:** Institute of Animal Science and Veterinary Medicine, Shandong Academy of Agricultural Sciences, Jinan Industry North Road 202, Jinan 250100, China

**Keywords:** COVID-19, antivirulent agents, bacteriophage, depolymerase

## Abstract

As of 29 July 2022, there had been a cumulative 572,239,451 confirmed cases of COVID-19 worldwide, including 6,390,401 fatalities. COVID-19 patients with severe symptoms are usually treated with a combination of virus- and drug-induced immuno-suppression medicines. Critical clinical complications of the respiratory system due to secondary bacterial infections (SBIs) could be the reason for the high mortality rate in COVID-19 patients. Unfortunately, antimicrobial resistance is increasing daily, and only a few options are available in our antimicrobial armory. Hence, alternative therapeutic options such as enzymes derived from bacteriophages can be considered for treating SBIs in COVID-19 patients. In particular, phage-derived depolymerases have high antivirulent potency that can efficiently degrade bacterial capsular polysaccharides, lipopolysaccharides, and exopolysaccharides. They have emerged as a promising class of new antibiotics and their therapeutic role for bacterial infections is already confirmed in animal models. This review provides an overview of the rising incidence of SBIs among COVID-19 patients. We present a practicable novel workflow for phage-derived depolymerases that can easily be adapted for treating SBIs in COVID-19 patients.

## 1. Introduction

The world has experienced four serious viral outbreaks during the last two decades including the 2002 Severe Acute Respiratory Syndrome (SARS) Coronavirus epidemic, the 2009 influenza A H1N1 pandemic, 2012 Middle East Respiratory Syndrome (MERS) outbreak, and recently the COVID-19 pandemic. In December 2019, Wuhan, China, experienced an outbreak of pneumonia with an unknown origin [1]. Different lethal novel strains spread in human populations frequently and increasing global concern. The earliest etiological research revealed that the SARS-CoV-2 virus was a member of the Coronaviridae family. Later, on 1 March 2020 the World Health Organization (WHO) declared it a pandemic on due to rapid and significant global propagation. COVID-19 illness is the biggest pandemic of our generation (WHO), with a cumulative total of 572,239,451 confirmed infected cases and 6,390,401 deaths (https://www.who.int/emergencies/diseases/novel-coronavirus-2019, accessed on 29 July 2022). The spectrum of clinical symptoms of COVID-19 is highly variable, range from mild respiratory disorders and fever to acute respiratory distress syndrome (ARDS), and along with other clinical complications, it can be fatal [2]. During any viral outbreaks, antiviral treatments are critical in order to reduce the morbidity and mortality rate. However, critically sick individuals may even have a high risk of developing secondary bacterial infections (SBIs) due to bacterial resistance to a combination of virus- and immuno-suppressive drugs [3]. Bacterial infections of secondary nature can occur during or after a viral illness when a person is exposed to a pathogen and has an immune system that cannot respond adequately [4]. SBIs are becoming more common in COVID-19 patients, increasing disease severity and mortality, particularly in those who require invasive mechanical ventilation [5]. At least one in seven COVID-19 patients was found to be additionally infected with a secondary bacterial infection with 50% of the fatalities during the SARS-CoV-2 pandemic caused by untreated or untreatable secondary bacterial infections, in most cases in the lung [6,7]. According to Elabbadi et al. (2021), severely ill COVID-19 patients have a very high rate of SBIs and are particularly susceptible to developing pneumonia afterward [8].

The recommendations to prevent disease transmission and treatment mainly focus on observed symptoms and the clinicians’ prescription. In the absence of effective antiviral medications, supportive management and next generation antibiotics can be the alternative therapeutic options to cope with symptoms of SBIs [4]. Unfortunately, antibiotics do not have impact on the virus itself and are being depleted to rescue COVID-19 patients from bacterial co-infections. Any surge in antibiotic use during the COVID-19 pandemic will have a detrimental effect on the prevalence of multidrug-resistant (MDR) bacteria and fuel global growth of antibiotic resistant bacterial pathogens [9]. In the past three years, the SARS-CoV-2 pandemic challenged the world and contributed further to the multidrug-resistance crises, due to the prophylactic administration of antibiotics in avoid to secondary infections [10]. The MDR becomes a worldwide public health issue since there are often no (chemical) antibiotics available, mainly for secondary infections [11]. Therefore, it is crucial to evaluate the morbidity caused by secondary infections and consider alternative clinical approaches to the currently used therapies [12].

Among some of the most promising approaches for the control of secondary bacterial infections is the use of phages and their related enzymes. Phages are found everywhere in the environment [13,14,15,16,17]. They are obligatory parasites and “natural enemies” of bacteria. The “predator-prey” interactions are identified as a potentially effective way to treat infections [18,19,20]. However, bacteria have evolved several defense mechanisms to prevent or lessen phage effects [21]. One method of phage resistance is to produce polysaccharide-thick layers, such as capsules, slime, or biofilm matrix, to prevent virion adsorption to the bacterial surface [22]. Bacteria use capsular polysaccharides (CPS) to defend against viral infection [23]. However, some phages evolved a particular technique to overcome this obstacle. Using virion-associated proteins with depolymerization activity, phages first recognize, bind to, and enzymatically destroy the CPS of the bacterial cell to gain access to the membrane and inject DNA. The tail fibers or tail spikes contain most of these structural enzymes, known as capsule depolymerases [24]. Phages now possess a wide range of bacteria-specific capsule depolymerases through the processes of natural selection [25].

The interest in anti-virulent agents, such as phage-encoded enzymes, has emerged as a promising method for infection management because of the increasing number of bacterial targets that might serve as the foundation for new antibiotic medication development [26]. Phage derived enzymes are more similar to conventional antibiotics and thus more suitable than whole phages for the current drug approval process. Phage-derived depolymerases administrated against bacterial capsules are showing therapeutic promise in animals against bacterial infections [27,28]. Capsule depolymerase aims to disarm the pathogen, in contrast to conventional antibiotics that either kill germs or stop their development [29]. In several animal, polymerases generated from phages that target bacterial capsules demonstrate high therapeutic potential. In some cases, enzymes may be preferred to complete phages since depolymerase does not lyse bacteria and does not emit endotoxin. The development and acceptance of phage-derived enzymes as a novel class of antibiotics could be substantially aided by in-depth knowledge of enzyme structure and dynamics [30].

In the subsequent sections of this opinion review, we will highlight the anti-virulence potential of phage-derived depolymerase enzymes against increasing rates of SBIs in COVID-19 patients. Additionally, we will propose a workflow of phage-derived enzymes that can be used to overcome the SBIs in COVID-19 patients.

## 2. Higher Rates of SBIs in COVID-19 Patients: A Primary Concern

COVID-19 has recently been identified as a life-threatening infectious disease, and scientists are trying to rapidly increase their understanding of the pathogenesis relevant to the disease [31]. High morbidity and mortality rates are primarily due to the prevalence of the SARS-CoV-2 coronavirus and subsequent microbial infections in the respiratory system [32]. Healthcare practitioners have raised concerns about secondary severe bacterial infection, and this concern has grown in the COVID-19 era. A link between COVID-19 infection and subsequent bacterial infections has already been established [33]. In COVID-19 patients, the rate of concurrent severe bacterial infections with viral disease is increasing and, consequently, so is extended hospitalization. This situation might worsen COVID-19 sickness and increase mortality [34].

*Acinetobacter baumanii*, *Pseudomonas aeruginosa*, *Klebsiella pneumoniae*, and *Staphylococcus aureus* are the most frequent pathogens found for SBIs in COVID-19 patients. The isolation ratio of carbapenem-resistant and colistin-resistant *A. baumannii*, *P. aeruginosa* and *K. pneumoniae* were 83.7%, 79.2%, 42.7%, and 5.6%, 1.7%, 42.7%, respectively [35]. *Acinetobacter pittii* clustering was seen in one of the ICUs in the hospital. This study also found an increasing frequency of multidrug resistant 92 (5.4%) *Corynebacterium striatum* isolates as a causative agent [35]. Due to *Enterococcus*, some COVID-19 patients had a higher incidence of BSI but not generally, and whole-genome sequencing of *Enterococcus* isolates demonstrated that nosocomial transmission did not explain the increased rate [36].

SBIs were found to be strongly associated with outcome severity in multicenter research involving 476 COVID-19 participants [2]. According to evidence from past pandemics and seasonal flu epidemics, co-infections may worsen viral diseases, but it is unclear if they definitively affect COVID-19 patient outcomes. Up to 30% of patients with secondary bacterial infections were found during the first SARS-CoV outbreak in 2003, and co-infection was positively related to disease severity [37]. In another study, bacterial co-infections are seen in 2–65% of patients during typical influenza seasons and are linked to increased morbidity and mortality [38]. The intensifying of bacterial co-infections during seasonal flu highlights the potential of studying the underlying phenomenon of pathogenicity, particularly with COVID-19.

A retrospective study that was published by Zhou et al., found that during the current COVID-19 pandemic, one in seven patients hospitalized with the illness developed a potentially fatal secondary bacterial infection, with nearly half of the non-survivors (27 out of 54) also creating a secondary infection and with ventilator-associated pneumonia developing in 10 of 32 patients (31%), necessitating invasive orienting mechanical ventilation [7]. A few studies indicated that COVID-19 patients have a more severe illness and a fatality rate three times higher than patients with influenza [39]. In addition, COVID-19 patients had two times the inpatient mortality rate from pulmonary secondary bacterial infections. Compared to influenza infection, COVID-19 patients required more time from admission to bacterial growth [40]. According to a meta-analysis of 24 cohort studies involving 3338 hospitalized COVID-19 patients, 3.5% of patients (with a 95% confidence interval (CI) of 0.4% to 6.7%) had bacterial co-infection at the time of presentation and 14.3% of patients (with a 95% CI of 9.6 to 18.9%) had secondary bacterial infection [41]. Based on the findings of microbial culture tests, 92 (8.7%) patients had respiratory or circulatory tract infections that were microbiologically confirmed. Out of 61 patients, respiratory tract infections were found in 44 patients to be monomicrobial and 17 patients to be polymicrobial [42]. Of 94 included patients, 68% acquired at least one of the studied SBIs during their ICU stay. Almost two-thirds of patients (65.96%, *n* = 62) attained secondary pneumonia [43].

COVID-19 patients had higher rates of bacterial infections (12.6% vs. 8.7%) and for a longer time (4 (1–8) vs. 1 (1–3) days) than other pneumonia patients. Gram-positive infections that developed later (> 48 h after admission) were more frequent in COVID-19 patients (28% vs. 9.5%). For COVID-19 patients, secondary infection was linked to a 2.7-fold increased risk of death [44]. According to Zhang et al., 22/38 patients (57.89%) experienced secondary infections. Secondary infections are more likely to occur in patients undergoing invasive mechanical ventilation or in critical condition (*p* < 0.0001). Lower discharge and increased mortality rates would result from secondary infections [45].

According to several disease severity markers, COVID-19 patients generally had more severe illnesses and worse outcomes, as indicated by a more significant percentage of intubations or deaths [46]. Importantly, COVID-19 patients had more secondary bacterial infections than had been described, which were separately linked to death in COVID-19. These data imply that SBIs may contribute significantly to disease severity in COVID-19 patients and may even be a therapeutic factor [10].

## 3. Major Challenges about SBIs Associated with COVID-19

Antimicrobial resistance (AMR) crises and prolonged hospitalization are major threats when we contact SBIs in the COVID-19 affected patients. Antimicrobials use is imperative for treating infectious diseases. The problem of antimicrobial resistance has worsened due to the indiscriminate use of antibiotics during the COVID-19 outbreak (Figure 1). Despite the widespread use of antibiotic therapy, the higher prevalence of SBIs in COVID-19 patients may be related to AMR bacteria in hospital settings [47]. The most prevalent infections discovered in blood and mucous samples of COVID-19 patients are *ESKAPE* pathogens (*Enterococcus faecium*, *S. aureus*, *K. pneumonia*, *A. baumannii*, *P. aeruginosa*, and *Enterobacter* species). During an influenza illness, secondary pneumonia is known to be caused by the *S. aureus* pathogens. Environmental modifications and immunological responses that produce favorable conditions for *S. aureus* infection are blamed for its dissemination to the lungs. Additionally, *A. baumannii* has been linked to long-term respiratory predisposition illnesses, such as influenza-like upper respiratory tract infections [48]. *P. aeruginosa* is a typical opportunistic pathogen of the respiratory system. Still, it is also recognized as the most prevalent Gram-negative bacterial species linked to serious hospital-acquired infections in several institutions [49]. Intensive care units (ICUs) and patients with impaired immune systems are particularly vulnerable to nosocomial infections caused by the Gram-negative, multiple-drug resistant (MDR) bacteria *A. baumannii* and *K. pneumoniae*. Among the microorganisms grown in blood cultures, coagulase-negative staphylococci with a percentage of 31%, and *A. baumannii* with 27.5% were prominent. In respiratory tract cultures, *A. baumannii* constitutes the majority with a rate of 33.3%, followed by *S. aureus* and *K. pneumonia* with a percentage of 9.5% each. The most resistant bacteria were *A. baumannii*, resistant to all antibiotics other than colistin [42]. There are not many antibiotic options available for those “superbugs” and to make matters worse, the use of some “last-resort” antibiotics, such as colistin, is closely regulated due to their potential for organ toxicity, disruption of normal flora, and AMR induction.

Several studies showed a longer hospitalization time for COVID-19 patients as compared to pneumonia with other pathogens. Therefore, critically ill people can easily acquire the SBIs during a more extended stay in the hospital. Moreover, a higher rate of bloodstream infections (BSIs) has been seen in COVID-19 patients given immunosuppressive treatments, e.g., tocilizumab, anakinra, and corticosteroids. As a result, mortality and admissions to ICU have increased in patients with BSI. Furthermore, COVID-19 infection induces pathological changes in the body such as a weakened immune system, diffuse alveolar damage, and dysregulated immune signaling. This situation leads to the commencement of SBIs and narrows down the potency of antibiotic treatments.

## 4. Phage-Derived Enzymes as an Alternative

Given the slow rate at which new antibiotics are being discovered, intact phages and their proteins are prospective treatments for bacteria [52,53,54], that are resistant to antibiotics [55]. The benefits of phage therapy include host specificity, amplification where bacteria are present in high concentrations and the evolving nature of phages that can counteract bacterial resistance [56]. Yet there are drawbacks, such as the need to match phages to the infecting strain and the simple fact that bacteria have many escape mechanisms. Furthermore, one significant factor preventing entire phages from being developed into drugs and approved is their complicated biology and pharmacological features. In this context, virus-encoded proteins with the potential to fight bacteria have attracted much research. Phage proteins can be extracted and also engineered for deployment as an alternative to whole phage therapeutic applications. These include virus-produced enzymes such as polysaccharide depolymerase, virion-associated lysins (VALs), and endolysins [50]. Endolysins are enzymes of a lytic nature employed by phages to break down bacterial peptidoglycan (PG) by the end of the replication cycle, leading to fast host lysis and the release of phage progeny [57]. At the start of infection, VALs and depolymerases help break down bacterial cell surface barriers by being attached to the virion particle [50]. In contrast to depolymerases, which break down polysaccharide molecules like capsules, lipopolysaccharide (LPS), or biofilm matrix, VALs are in charge of breaking down PG needed for injection of phage genetic material into the infected host cell [24].

## 5. Exploring the Antivirulence Potential of Phage-Derived Enzymes: Depolymerases

The bulk of phage polysaccharide depolymerases are linked to the virion surface and are frequently encoded as parts of structural proteins such as tail fibers. Depolymerases are believed to act on the CPS, exopolysaccharide (EPS), or LPS of their host bacterium, cleaving these polymeric compounds produced by the host cell and exposing the cell surface receptors required for binding, facilitating the phage infection (Figure 2). Depolymerases are now being researched to prevent and treat biofilm-related infections because CPS, EPS, and LPS play a significant role in forming biofilms. Phage depolymerases’ manner of action is their principal benefit. Depolymerases, non-lytic enzymes, function as antivirulent agents, reducing the severity of the infection and assisting the host’s immune system in removing the infection. Despite these appealing characteristics, the majority of the research into the therapeutic use of phage depolymerases has so far been concentrated on a small number of bacterial species where the critical role of capsular polysaccharides as virulence factors has been well established, particularly in *ESKAPE* pathogens [15,16,17]. Depolymerase enzymes are now being investigated as prospective antimicrobials for the treatment of biofilm-related infections due to their capacity to both prevent the production of new biofilms and break down existing ones. According to some findings, depolymerases can be used as an adjuvant antibiotic to fight MDR microorganisms and support the development of novel antibacterial drugs. These outcomes suggest that these phage-derived enzymes could be a game-changer for treating secondary infections in COVID-19 patients who are also resistant to treatment.

Capsular depolymerases show an interesting type of antibiotic: they do not kill, but merely strip the bacteria of protective polysaccharides and thus expose the bacteria to immune components [27]. They have a potential advantage over endolysins in that they do not lyse the bacteria, thereby minimizing inflammatory responses from endotoxins [58]. In vivo studies of capsular depolymerases are limited as yet but their effectiveness has been demonstrated in animal models (Table 1). It has been shown that recombinant depolymerases shield mice against lethal systemic bacterial infections [50,59,60], and dislodge biofilms to improve antibacterial efficacy [61]. Depolymerases and antibiotics administered together will enhance antibacterial efficacy; this is predicted but needs to be adequately validated by trials. The adjuvant effect of gentamicin and a depolymerase produced from *Aeromonas punctata* (a facultative anaerobic Gram-negative bacterium) in treating mice infected with non-lethal doses of *K. pneumoniae* was first documented [62]. For systemic infection and lung infection, respectively, intravenous administration of the combination dramatically decreased bacterial counts compared to single-agent therapies. They explained the increased bacterial susceptibility to gentamicin after the depolymerase decapsulated the germs as the cause of the increased bacterial killing efficiency. To help gentamicin penetrate *K. pneumoniae* biofilms, depolymerases also successfully dispersed the EPS matrix [63]. The polymyxin B and Dep42 depolymerase work synergistically when treating *K. pneumoniae* biofilms [64]. Contrarily, Latka and Drulis-Kawa demonstrated that the KP34p57 depolymerase was not having any effect on the action of ciprofloxacin but might significantly increase the antibiofilm effectiveness of phages that do not produce depolymerase [65]. Depolymerase may improve the antibiofilm performance of the antibacterial enzyme endolysin, which is encoded by a phage [66].

## 6. Challenges to Consider in COVID-19 SBIs

Phages are biological entities, so phage-based products should be manufactured using methods based on good manufacturing practices (GMP) [73]. The quality control of phage-based products is another significant consideration. They should be checked frequently for stability (shelf life), sterility, cytotoxicity, and periodic pH readings [74]. Phage-based products usually raise the concern that widespread usage of these products could lead to a problem akin to antibiotic resistance [56]. Some additional limitations can highlight the scarcity of phage-derived enzyme practice for COVID-19 patients.

First, SARS-CoV-2 infection induced some pathological changes in the body, i.e., a secondary pulmonary disease that can create blockage in the airway passage, which is a major obstacle in the delivery of therapeutic enzymes at the bacterial infection site. Second, effective enzymes based therapies always involve the patient’s immune responses. Regrettably, COVID-19 may affect immune signaling and damage the immune cells at the bacterial infection site. Moreover, it is observed that immuno-suppressive therapies have been employed in COVID-19 patients, which leads to BSIs [75]. This alarming situation raises more challenges and obstacles for exterminating resistant bacteria. It leads to the evolution of phage-resistant bacteria, a critical characteristic of phage-based enzymes for SBIs related to COVID-19. Third, SARS-CoV-2 is very dangerous and its management requires a biosafety level-3 (BSL-3) lab and trained staff with proper equipment, which increases the cost and complications and makes it laborious. Moreover, after infection, it spreads all over the body, damaging many organs and complicating the enzyme-based therapy process, including phage isolation, screening, protein cloning, expression, purification, and evaluation of its efficacy.

Patient care and bacterial sample collection are routine processes performed in specified patient wards and laboratories by trained staff equipped with a BSL-3 lab. Ready-to-use vials are routinely formulated in standard labs using specific host bacteria, and a GMP approved plant was used for their containment. On the other hand, a customized therapy based on a phage product requires a stranded phage library which is then passed on to a designated BSL-2 lab for phage screening, and protein purification evaluation under PPE conditions. The purified protein is transferred to the BSL-3 lab for inspection of efficiency. Finally, proficient protein vials that contain highly tittered protein against specific bacteria are transported for enzyme therapy.

## 7. A Practicable Workflow

A procedure that will depend on the cooperation of many functional areas calls for different PPE levels. In a hospital that has been COVID-19 designated, standard patient care and bacterial culture will probably be carried out in the inpatient ward and clinical laboratory. A dedicated portion of the clinical laboratory will be specified for phage screening and efficiency analysis under BSL-3 lab PPE circumstances. Phages will be routinely amplified by growing in the original host bacterium in the typical microbiology laboratory to create ready-to-use phage vials. A packaging facility with GMP certification will pack the vials. Qualified vials containing purified protein can be quickly chosen and delivered to the inpatient ward for therapy by utilizing the material flow (from lower BSL lab zones to higher BSL lab zones) and the reverse information flow. Bacterial isolates will frequently be phage-typed for epidemiological purposes. This information will ultimately help create enough phage-derived therapeutic proteins and combined broad-spectrum, fixed-composition cocktails for emergencies (Figure 3).

## 8. Conclusions

The end of the COVID-19 pandemic will take a long time, even though massive efforts have been made to control it. Numerous disease severity markers showed that some COVID-19 individuals are more seriously unwell and have worse outcomes. Our ability to remove MDR bacteria is waning as they become more prevalent, which worsens the SBIs in COVID-19 patients. Although there have been few clinical studies on SBIs in COVID-19, the results show that the illness may be treatable. Alternative phage-based treatments can be utilized when the complete phage may not be as effective due to problems including resistance, host specificity and drug development process through the purification and characterization of phage-derived antimicrobials. Using enzymes produced from phages slows down resistance development dramatically. However, some limitations and important questions arise about the delivery route of these phage-derived therapeutic and host immune responses.

## Figures and Tables

**Figure 1 microorganisms-11-00424-f001:**
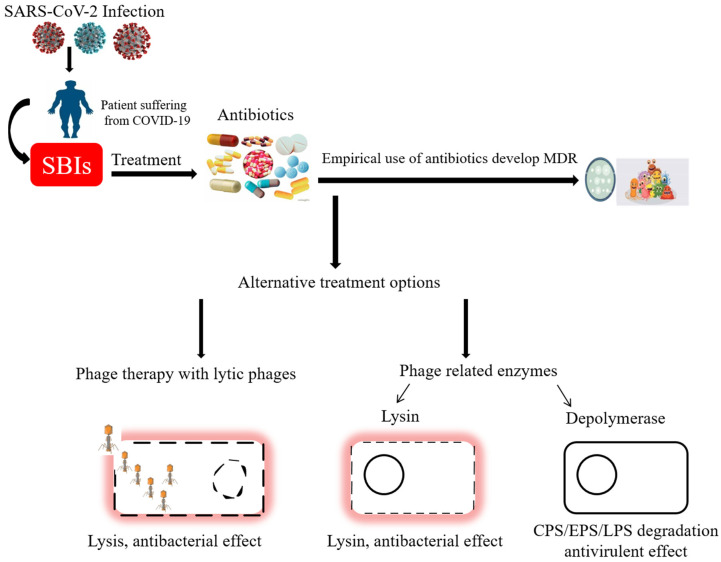
Schematic representation of drug-resistant secondary infections in SARS-CoV-2-infected patients and possible alternative treatments [50,51].

**Figure 2 microorganisms-11-00424-f002:**
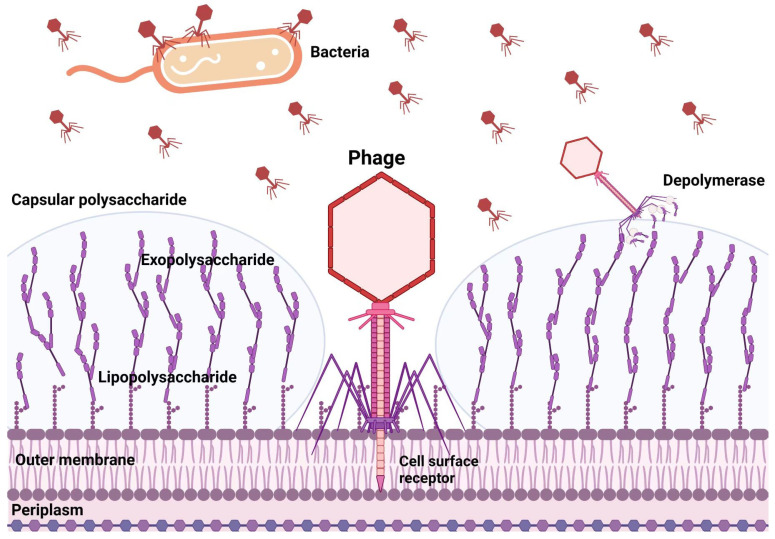
Polysaccharide depolymerase of bacteriophages attacks on different targets.

**Figure 3 microorganisms-11-00424-f003:**
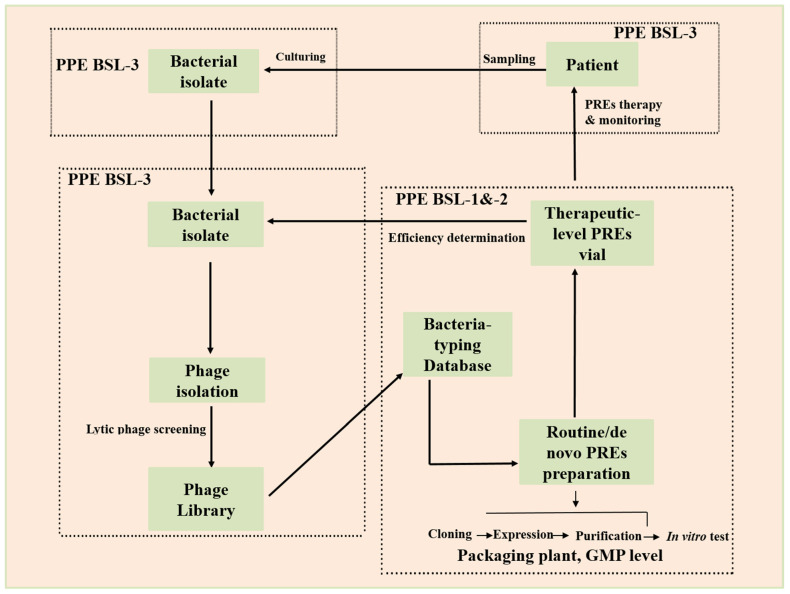
Workflow for COVID-19 infection management proposed in this study [76]. With an established phage library, against a given pathogenic bacterium that has been isolated from a patient or a hospital surface are screened, and pre-stocked phage-enzymes vials are delivered for application to the patient or the environment. PREs: phage-related enzymes.

**Table 1 microorganisms-11-00424-t001:** The application of phage-derived depolymerases in animal models.

Phage	Enzyme	Model	Delivery Route	Outcome	Reference
*Escherichia coli* K1	EndoE endosialidase from Coliphage E	Neonatal rat model of bacteremia	Intraperitoneal injection	100% of animals protected from death	[67]
*Salmonella Typhimurium*	P22sTsp endorhamnosidase from *Salmonella* phage P22	Chicken model of gastrointestinal infection	Oral administration	Bacterial cfu reduction of ~1 order	[68]
*Klebsiella pneumoniae*	K64dep capsule depolymerase from *Klebsiella* phage K64-1	Mouse model of bacteremia	Intraperitoneal injection	100% of animals protected from death	[69]
*Pseudomonas aeruginosa*	LKA1gp49 LPS lyase from *Pseudomonas* phage LKA1	*Galleria mellonella* infection model	Injection into the last pro-leg	20% of animals protected from death	[21]
*Klebsiella pneumoniae*	Dep_kpv79 and Dep_kpv767 depolymerase	Mouse model	Intraperitoneal injection, Intramuscular injection	80%, 100%	[70]
*Acinetobacter* *baumannii*	Depolymerase Dpo71	*Galleria mellonella* infection model	Injection into the last pro-leg	80%	[61]
*Acinetobacter* *baumannii*	Capsule depolymerase B9gp69	Cell line model of human lung	-	-	[71]
*Proteus mirabilis*	Phage derived-Depolymerase	*Galleria mellonella* infection model	Injection into the last pro-leg	20% protected from death	[72]
*Escherichia coli*	O91-specific polysaccharide depolymerase	Mouse model	Injection into the last pro-leg	83% survival	[60]

## Data Availability

Not applicable.

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
