# Peer review of "Phage-Derived Depolymerase: Its Possible Role for Secondary Bacterial Infections in COVID-19 Patients"

_microorganisms, 2023, doi:10.3390/microorganisms11020424_

Round 1
Reviewer 1 Report
The manuscript “Phage-derived depolymerase: the possible role for secondary bacterial infections in COVID-19 patients” by Song et al. makes a case for using phage-derived enzymes to combat secondary bacterial infections (SBIs) in COVID-19 patients. In general, I am not sure what the manuscript adds to the current body of literature except it is a slightly different angle on phage therapy in COVID-19 SBIs compared to what is already published.
Major points:
1. I am unconvinced by the framing of this review. Many reviews on phage therapy, through whole phage or phage enzymes, have been published. It is unclear how adding that these are SBIs of COVID-19 specifically contributes to the body of work.
2. I am not sure how this workflow is specific to COVID-19 aside from the need for BSL-3 space for part of it. The idea is not uncommon and is already presented in Wu et al. 2022 (doi: 10.1016/j.coviro.2021.11.001).
3. Figure 3 is very similar to Figure 1 in Wu et al. 2022, though the focus is on whole phage instead of depolymerases.
4. The fact that Wu et al. 2022 is not cited is highly suspicious.
Minor points:
1. In general, there are numerous formatting issues, including:
- missing spaces between words: e.g. lines 81 and 82, “toICUhaveincreased” line 174;
- missing italics: e.g. “The Lancet” in line 104, “Aeromonas punctata” in lines 226-227;
- word choice throughout the paper seems overly casual or colloquial
2. Line 57: Bacteria are not really phage food in the sense that phage cannot consume and metabolize bacteria.
3. Line 67: “thanks to evolutionary development” should be changed to “through the processes of natural selection” or similar.
4. Line 182: “a wealth” of what?
5. Figure 3: What is/are “PREs”?
Author Response
We would like to express our sincere thank your helpful comments and suggestions, which allowed us to further improve our manuscript. This manuscript has been carefully modified according to the comments. The following response is point-by-point towards the reviewers’ comments.
The manuscript “Phage-derived depolymerase: the possible role for secondary bacterial infections in COVID-19 patients” by Song et al. makes a case for using phage-derived enzymes to combat secondary bacterial infections (SBIs) in COVID-19 patients. In general, I am not sure what the manuscript adds to the current body of literature except it is a slightly different angle on phage therapy in COVID-19 SBIs compared to what is already published.
Major points:
I am unconvinced by the framing of this review. Many reviews on phage therapy, through whole phage or phage enzymes, have been published. It is unclear how adding that these are SBIs of COVID-19 specifically contributes to the body of work.
The mortality and morbidity rate due to secondary infections associated with COVID-19 is immense. The impractical use of antibiotics has laid down a perfect foundation for the development of drug-resistant pathogens. Our narrative highlights an effective therapy using bacteriophage related enzyme depolymerase against these drug-resistant pathogens related to COVID-19. This different perspective of the treatment of resistant pathogens is a bypass for the way we see medicine.
I am not sure how this workflow is specific to COVID-19 aside from the need for BSL-3 space for part of it. The idea is not uncommon and is already presented in Wu et al. 2022 (doi: 10.1016/j.coviro.2021.11.001).
SARS-CoV-2 pneumonia is associated with a longer duration of illness than pneumonia attributed to other pathogens. As COVID-19 is life threatening, so we need to follow a workflow involving BSL-3 lab. We emphasize on a effective therapy using bacteriophage related enzyme depolymerase against these drug-resistant pathogens associated with COVID-19. Phage therapy has still facing several hurdles including safety, efficacy, accessibility, acceptability and regulatory issues. Concerning the application of phage-based depolymerase, the preliminary studies involving animal models and clinical trials are demonstrating promising antibacterial efficacy and confirming their safety.
Figure 3 is very similar to Figure 1 in Wu et al. 2022, though the focus is on whole phage instead of depolymerases.
We need to do some more procedures during de. novo preparations as we need to do clone, pure and in vitro test of enzyme. We also cited the paper Wu et al. 2022.
The fact that Wu et al. 2022 is not cited is highly suspicious.
Dear sir, we have incorporated the reference in revised version of the manuscript.
Minor points:
In general, there are numerous formatting issues, including:
- missing spaces between words: e.g. lines 81 and 82, “toICUhaveincreased” line 174;
We are thankful for your suggestion. We have corrected the mistakes.
- missing italics: e.g. “The Lancet” in line 104, “Aeromonas punctata” in lines 226-227;
We have corrected according to the suggestion.
- word choice throughout the paper seems overly casual or colloquial
English improves throughout the paper.
Line 57: Bacteria are not really phage food in the sense that phage cannot consume and metabolize bacteria.
Thank you for this valuable feedback .We have makes changes accordingly.
Line 67: “thanks to evolutionary development” should be changed to “through the processes of natural selection” or similar.
We appreciate the valuable advice from the reviewer. This line has been rephrased according to your suggestion.
Line 182: “a wealth” of what?
We are very thankful for highlighting the mistake. We have rephrased the sentence.
Figure 3: What is/are “PREs”?
The abbreviation has been mentioned in the paper.

Reviewer 2 Report
Peer reviewed work
microorganisms-2157532 is, in my opinion, of considerable interest. The high risk of bacterial complications associated with COVID-19 may be one of the main causes of death. The approach proposed by the authors to the use of depolymerase as an important component of SBIs therapy may be accompanied by an increase in the effectiveness of therapy for complicated COVID-19. The approach proposed by the authors is convincingly illustrated by the cited literature. I have no significant comments on the work. I believe that the peer-reviewed review responds to the requests of the journal and may have been published in the proposed edition.
Author Response
We thank the reviewer for the positive comment.

Reviewer 3 Report
The manuscript “Phage-derived depolymerase: the possible role for secondary bacterial infections in COVID-19 patients” by Song et al., provides information about the role of secondary bacterial infections (SBI) for COVID-19 patients and highlights the putative role of phage-derived depolymerases used to treat SBIs in COVID-19 patients.
Major points:
I do not understand why the authors do not provide sufficient literature references. In my opinion, there are too many passages without any reference! See some examples below.
Literature missing:
Line 42-44: SBIs are becoming more common in COVID-19 patients, leading to increased disease severity and mortality, particularly in those who require invasive mechanical ventilation.
Line 74-77: In other cases, enzymes may be preferable than complete phages since depolymerase does not lyse bacteria and does not emit endotoxin as a result. The development and acceptance of phage-derived enzymes as a novel class of antibiotics could be substantially aided by a greater knowledge of enzyme structure and enzyme dynamics.
Paragraph 3: For a review, I feel like one reference is to less for a complete paragraph. I would appreciate if the authors would provide the reader with more references.
Paragraph 4: Same as for paragraph 3. No references from line 183 to line196!!!
Line: 256-257: Moreover, it is observed that immuno-suppressive therapies have been employed at COVID-19 patients which leads to BSIs.
Minor points:
Line 38: COVID-19 no parentheses
Line 45: Should be [3] instead of [11]
Line 57: …, because they are phages´ natural food,…
I would change the word food because it is colloquial and misleading or at least put it in quotation marks.
Line 67: depolymerases instead of depolymerase
Line 79-82, 169-175: In this sections are several spaces missing.
Line 99-130: This section is quit inconsistent concerning the format of numbers and units, etc.
For example: Line 100: only time that percent is written and not % used
Line 108: ten instead of 10
Line 126: COVID19 without hyphen
Line 127: I do not understand the meaning of the numbers (1.22-58.3)
Line 146: I would suggest to reorder the names of the bacteria, that it fits with the word ESKAPE
Line149/151: Bacterial names are abbreviated but in lines 146-147 they are fully written
Line 226: Aeromonaspunctata not italic, no space
Author Response
We would like to express our sincere thank your helpful comments and suggestions, which allowed us to further improve our manuscript. This manuscript has been carefully modified according to the comments. The following response is point-by-point towards the reviewers’ comments.
The manuscript “Phage-derived depolymerase: the possible role for secondary bacterial infections in COVID-19 patients” by Song et al., provides information about the role of secondary bacterial infections (SBI) for COVID-19 patients and highlights the putative role of phage-derived depolymerases used to treat SBIs in COVID-19 patients.
Major points:
I do not understand why the authors do not provide sufficient literature references. In my opinion, there are too many passages without any reference! See some examples below.
Dear sir, we have incorporated enough references as per your suggestion.
Literature missing:
Line 42-44: SBIs are becoming more common in COVID-19 patients, leading to increased disease severity and mortality, particularly in those who require invasive mechanical ventilation.
We are obliged for your suggestion; it has been incorporated in the updated version of manuscript.
Line 74-77: In other cases, enzymes may be preferable than complete phages since depolymerase does not lyse bacteria and does not emit endotoxin as a result. The development and acceptance of phage-derived enzymes as a novel class of antibiotics could be substantially aided by a greater knowledge of enzyme structure and enzyme dynamics.
Manuscript updated according to your suggestion.
Paragraph 3: For a review, I feel like one reference is to less for a complete paragraph. I would appreciate if the authors would provide the reader with more references.
We have added more references accordingly.
Paragraph 4: Same as for paragraph 3. No references from line 183 to line196!!!
References has been incorporated accordingly.
Line: 256-257: Moreover, it is observed that immuno-suppressive therapies have been employed at COVID-19 patients which leads to BSIs.
Reference has been included according to your suggestion.
Minor points:
Line 38: COVID-19 no parentheses
Revised as per your suggestion.
Line 45: Should be [3] instead of [11]
Revised as per your suggestion.
Line 57: …, because they are phages´ natural food,…
This line has been rephrased accordingly.
I would change the word food because it is colloquial and misleading or at least put it in quotation marks.
This line rephrased and made correction according to your suggestion.
Line 67: depolymerases instead of depolymerase
This mistake is corrected according to your suggestion.
Line 79-82, 169-175: In this sections are several spaces missing.
All corrections have been done related to space in both sections.
Line 99-130: This section is quit inconsistent concerning the format of numbers and units, etc.
For example: Line 100: only time that percent is written and not % used
Line 108: ten instead of 10
Line 126: COVID19 without hyphen
Manuscript has been updated according to the recommendations.
Line 127: I do not understand the meaning of the numbers (1.22-58.3)
We have removed these number in order to avoid the confusion.
Line 146: I would suggest to reorder the names of the bacteria, that it fits with the word ESKAPE
We have make changes as per your suggestion.
Line149/151: Bacterial names are abbreviated but in lines 146-147 they are fully written
We have made corrections accordingly.
Line 226: Aeromonaspunctata not italic, no space
Respected reviewer, We have make changes accordingly.
- Changes are highlighted in the revised version of manuscript.

Round 2
Reviewer 3 Report
I would like to thank the authors for changing the manuscript according to my suggestions.
Author Response
Thank you!